# Integral-Related ZE (Zhang-Equivalency) Equations and Inequations of Unequal-Parameter-Value (UPV) Situation

Yunong Zhang, Jiale Zhang, Peng An

School of Computer Science and Engineering, Sun Yat-sen University, Guangzhou 510006, P. R. China
E-mail: zhynong@mail.sysu.edu.cn, zhangjle9@mail2.sysu.edu.cn, anpeng3@mail2.sysu.edu.cn

Abstract: In this paper, we elaborate on the notion of Zhang equivalency (ZE) and subsequently explore the scenarios of equations and inequations within the context of the unequal-parameter-value (UPV) situation. Utilizing the framework of ZE, we conduct a thorough examination of the equations and inequations pertaining to once, twice, thrice, and $n$-times. Futhermore, we arrive at the general integral-related ZE formulas.

Key Words: Zhang equivalency, Unequal-parameter-value, Inequation, Time-varying problems solving.

## 1 Introduction

Equivalence and its specific forms of equivalence serve as effective and robust tools in addressing a multitude of issues across various disciplines, including mathematics, physics, and engineering (encompassing computation). Broadly, equivalence refers to the process of transforming a given problem into an equivalent one that is either easier to solve or more readily understood. Zhang equivalency has introduced integral forms, equations, and inequations under conditions of equivalent parameters, which were discussed in [1]-[6]. Building on this groundwork, we intend to provide a supplement by generalizing these concepts to scenarios under unequal parameter conditions.

In this paper, the mainly contributions of this paper are listed as follows.

1) Introduction of standard ZE equations and inequations of equal-parameter-value (EPV).
2) Derivation of standard ZE equations and inequations of unequal-parameter-value (UPV).
3) Derivation of once-integral ZE equations and inequations of UPV.
4) Derivation of twice-integral ZE equations and inequations of UPV.
5) Derivation of thrice-integral ZE equations and inequations of UPV.
6) Derivation of $n$-times-integral ZE equations and inequations of UPV.

## 2 Standard ZE Equations and Inequations of Equal-Parameter-Value (EPV) Situation

In this section, the standard ZE equations and the standard ZE inequations of EPV situation are discussed, which are proposed in [2] and [5].

### 2.1 Standard ZE Equations of EPV Situation

In the following section, we specially discuss the formulas for the 0-order, 1-order, 2-order, 3-order, and $n$-order, which are proposed in [2] and [5].

1) Order-0 ZE General Formula of EPV Situation: Based on ZE, we propose an error function denoted

by $e(t) = a(t) - d(t)$, which is defined as the difference between the desired value $a(t)$ and the actual value $d(t)$. It can be inferred that the equation $e(t) = 0$ holds true, as $t \to \infty$.

$$e(t) = 0. \tag{1}$$

2) Order-1 ZE General Formula of EPV Situation:

$$\dot{e}(t) + \lambda e(t) = 0, \tag{2}$$
$$\dot{a}(t) - \dot{d}(t) + \lambda(z(t) - z_{\mathrm{d}}(t)) = 0. \tag{3}$$

In the present context, let $\dot{e}(t)$ denote the first-order reciprocal $\mathrm{d}(e(t))/\mathrm{d}t$, and let $\lambda \gg 0$. Based on ZE, it can be demonstrated that the (1), (2), and (3) are equivalent.

3) Order-2 ZE General Formula of EPV Situation:

$$\ddot{e}(t) + 2\lambda \dot{e}(t) + \lambda^2 e(t) = 0. \tag{4}$$

According to ZE, since (1) is equivalent to (2), we define $\epsilon_1(t) = \dot{e}(t) + \lambda e(t)$ and then substitute it into $\dot{\epsilon}_1(t) + \lambda \epsilon_1(t) = 0$. After some straightforward simplification, we arrive at (4).

4) Order-3 ZE General Formula of EPV Situation:

$$\dddot{e}(t) + 3\lambda \ddot{e}(t) + 3\lambda^2 \dot{e}(t) + \lambda^3 e(t) = 0. \tag{5}$$

Let $\epsilon_2(t) = \ddot{e}(t) + 2\lambda \dot{e}(t) + \lambda^2 e(t)$ and then substitute it into $\dot{\epsilon}_2(t) + \lambda \epsilon_2(t) = 0$. We obtain the formula through straightforward simplification as $t \to \infty$ and $\lambda \gg 0$.

5) Order-$n$ ZE General Formula of EPV Situation:

$$\binom{n}{0} e^{(n)}(t) + \binom{n}{1} \lambda e^{(n-1)}(t) + \ldots + \binom{n}{k} \lambda^k$$
$$e^{(n-k)}(t) + \ldots + \binom{n}{n-1} \lambda^{(n-1)} \dot{e}(t) + \tag{6}$$
$$\binom{n}{n} \lambda^n e(t) = 0.$$

We proceed by employing the method of mathematical induction. Suppose that the formula for the $(n-1)$th order is expressed as $\epsilon_{n-1}(t)$. Moreover, it has been established that the formula is

This work is supported by the National Natural Science Foundation of China (with number 62376290). Besides, the corresponding author is Yunong Zhang.

congruent with (1) as $t \to \infty$ and $\lambda \gg 0$. Subsequently, by incorporating this into (2) and performing a series of straightforward algebraic reductions, we are able to arrive at (6). Besides, when $\lambda_1 \gg 0$, $\lambda_2 \gg 0$, ..., $\lambda_n \gg 0$ and $t \to \infty$, the derived equations are equivalent.

$$
\begin{cases}
e(t) = 0, \\
\dot{e}(t) + \lambda e(t) = 0, \\
\ddot{e}(t) + 2\lambda \dot{e}(t) + \lambda^2 e(t) = 0, \\
\dddot{e}(t) + 3\lambda \ddot{e}(t) + 3\lambda^2 \dot{e}(t) + \lambda^3 e(t) = 0, \\
\binom{n}{0} e^{(n)}(t) + \binom{n}{1}\lambda e^{(n-1)}(t) + \ldots + \\
\binom{n}{k}\lambda^k e^{(n-k)}(t) + \ldots + \binom{n}{n-1}\lambda^{(n-1)}\dot{e}(t) + \\
\binom{n}{n}\lambda^n e(t) = 0.
\end{cases}
$$

## 2.2 Standard ZE Inequations of EPV Situation

In the following section, we specially examine the formulas for the 0-order, 1-order, 2-order, 3-order, and $n$-order.

1) Order-0 ZE General Formula of EPV Situation:
   Based on ZE, the derivation of the subsequent formula is straightforward as $t \to \infty$.

$$
e(t) \le 0. \tag{7}
$$

2) Order-1 ZE General Formula of EPV Situation:

$$
\dot{e}(t) + \lambda e(t) \le 0, \tag{8}
$$

$$
\dot{a}(t) - \dot{d}(t) + \lambda(a(t) - d(t)) \le 0. \tag{9}
$$

According to ZE, we arrive at (8) and (9).

3) Order-2 ZE General Formula of EPV Situation:

$$
\ddot{e}(t) + 2\lambda \dot{e}(t) + \lambda^2 e(t) \le 0. \tag{10}
$$

As indicated by ZE, (7) and (8) are considered equivalent as $t \to \infty$ and $\lambda \le 0$. Proceeding with the definition of $\epsilon_1(t) = \dot{e}(t) + \lambda e(t)$, we substitute this expression into $\dot{\epsilon}_1(t) + \lambda\epsilon_1(t) \le 0$. Through simplification, we subsequently arrive at (10).

4) Order-3 ZE General Formula of EPV Situation:

$$
\dddot{e}(t) + 3\lambda \ddot{e}(t) + 3\lambda^2 \dot{e}(t) + \lambda^3 e(t) \le 0. \tag{11}
$$

Based on ZE, it is evident that (7) and (10) are equivalent as $t \to \infty$ and $\lambda \gg 0$. In light of this, we define $\epsilon_2(t) = \ddot{e}(t) + 2\lambda\dot{e}(t) + \lambda^2 e(t)$ and proceed to incorporate this substitution into $\dot{\epsilon}_2(t) + \lambda\epsilon_2(t) \le 0$. Following a process of simplification, we arrive at (11).

5) Order-$n$ ZE General Formula of EPV Situation:

$$
\binom{n}{0} e^{(n)}(t) + \binom{n}{1}\lambda e^{(n-1)}(t) + \ldots + \binom{n}{k}\lambda^k
$$
$$
e^{(n-k)}(t) + \ldots + \binom{n}{n-1}\lambda^{(n-1)}\dot{e}(t) +
$$
$$
\binom{n}{n}\lambda^n e(t) \le 0. \tag{12}
$$

The proof procedure is largely analogous to (6). By employing mathematical induction, we proceed on the assumption that the formula of the $(n-1)$th order is valid. We then define

$$
\epsilon_{n-1} = \binom{n-1}{0}e^{(n-1)}(t) + \binom{n-1}{1}\lambda e^{(n-2)}(t) +
$$
$$
\ldots + \binom{n-1}{k}\lambda^k e^{(n-1-k)}(t) + \ldots +
$$
$$
\binom{n-1}{n-2}\lambda^{(n-2)}\dot{e}(t) + \binom{n-1}{n-1}\lambda^{n-1}e(t)
$$

and incorporate this into (8). Through this substitution, we are able to obtain (12). When $\lambda_1 \gg 0$, $\lambda_2 \gg 0$, ..., $\lambda_n \gg 0$ and $t \to \infty$, the derived inequations are equivalent.

$$
\begin{cases}
e(t) \le 0, \\
\dot{e}(t) + \lambda e(t) \le 0, \\
\ddot{e}(t) + 2\lambda \dot{e}(t) + \lambda^2 e(t) \le 0, \\
\dddot{e}(t) + 3\lambda \ddot{e}(t) + 3\lambda^2 \dot{e}(t) + \lambda^3 e(t) \le 0, \\
\binom{n}{0} e^{(n)}(t) + \binom{n}{1}\lambda e^{(n-1)}(t) + \ldots + \\
\binom{n}{k}\lambda^k e^{(n-k)}(t) + \ldots + \binom{n}{n-1}\lambda^{(n-1)}\dot{e}(t) + \\
\binom{n}{n}\lambda^n e(t) \le 0.
\end{cases}
$$

## 3 Standard ZE Equations and Inequations of Unequal-Parameter-Value (UPV) Situation

In this section, we proceed to discuss the standard ZE equations and inequations in unequal-parameter-value.

### 3.1 Standard ZE Equations of UPV Situation

In this section, we specially discuss the equations for the 0-order, 1-order, 2-order, 3-order, and $n$-order.

1) Order-0 ZE General Formula of UPV Situation:
   In analogy to the previously established 0th-order formula, the following relationship is maintained as $t \to \infty$.

$$
e(t) = 0. \tag{13}
$$

2) Order-1 ZE General Formula of UPV Situation:
   The order-1 ZE general formula of equation with $t \to \infty$ and $\lambda_1 \gg 0$ is presented as

$$
\dot{e}(t) + \lambda_1 e(t) = 0, \tag{14}
$$

$$
\dot{a}(t) - \dot{d}(t) + \lambda_1(a(t) - d(t)) = 0. \tag{15}
$$

3) Order-2 ZE General Formula of UPV Situation:

$$
\ddot{e}(t) + (\lambda_1 + \lambda_2)\dot{e}(t) + \lambda_1\lambda_2 e(t) = 0. \tag{16}
$$

Similarly, we define $\epsilon_1(t)$ as $\dot{e}(t) + \lambda_1 e(t)$. By incorporating into (14), we can derive the following equation.

$$
\dot{\epsilon}_1(t) + \lambda_2\epsilon_1(t) = 0.
$$

Following simplification, we could arrive at (16).

4) Order-3 ZE General Formula of UPV Situation:

$$
\dddot{e}(t) + (\lambda_1 + \lambda_2 + \lambda_3)\ddot{e}(t) + (\lambda_1\lambda_2 + \lambda_1\lambda_3 + \lambda_2\lambda_3)\dot{e}(t) + \lambda_1\lambda_2\lambda_3 e(t) = 0. \tag{17}
$$

Similarly, we can also set $\epsilon_3(t)$ as $\ddot{e}(t) + (\lambda_1 + \lambda_2)\dot{e}(t) + \lambda_1\lambda_2 e(t)$ and then substitute it into $\dot{\epsilon}_3(t) + \lambda_3\epsilon_3(t) = 0$. After simplification, it is not difficult to derive (17).

5) Order-$n$ ZE General Formula of UPV Situation:

$$e^{(n)}(t) + \sum_{i=1}^{n-1} e^{(i)}(t)\bigg(\sum_{j=1}^{n}\sum_{z=j+1}^{n}\cdots\sum_{w=n-i}^{n}\lambda_j\lambda_z$$
$$\cdots\lambda_w\bigg) + \lambda_1\ldots\lambda_n e(t) = 0. \tag{18}$$

Assuming formula is true for 1 to $n$. Then we define (18) as $\epsilon_n(t)$. Through the process of simplification, we obtain the subsequent mathematical expression:

$$\dot{\epsilon}_n(t) + \lambda_{n+1}\epsilon_n(t) = 0.$$

$$e^{(n+1)}(t) + \sum_{i=1}^{n-1} e^{(i+1)}(t)\bigg(\sum_{j=1}^{n}\sum_{z=j+1}^{n}\cdots\sum_{w=n-i}^{n}\lambda_j\lambda_z$$
$$\cdots\lambda_w\bigg) + \lambda_1\ldots\lambda_n e^{(1)}(t) + \lambda_{n+1}\bigg(e^{(n)}(t) +$$
$$\sum_{i=1}^{n-1} e^{(i)}(t)\bigg(\sum_{j=1}^{n}\sum_{z=j+1}^{n}\cdots\sum_{w=n-i}^{n}\lambda_j\lambda_z\ldots\lambda_w\bigg)$$
$$+\lambda_1\ldots\lambda_n e(t)\bigg) = 0. \tag{19}$$

Through a straightforward simplification process, we arrive at the following formula.

$$e^{(n+1)}(t) + \sum_{i=1}^{n} e^{(i)}(t)\bigg(\sum_{j=1}^{n+1}\sum_{z=j+1}^{n+1}\cdots\sum_{w=n+1-i}^{n+1}\lambda_j\lambda_z$$
$$\cdots\lambda_w\bigg) + \lambda_1\ldots\lambda_{n+1}e(t) = 0.$$

Therefore, we conclude that the formula for the $(n+1)$ th order is valid. When $\lambda_1 \gg 0$, $\lambda_2 \gg 0$, $\ldots, \lambda_n \gg 0$ and $t \to \infty$, the derived equations are equivalent.

$$\begin{cases} e(t) = 0, \\ \dot{e}(t) + \lambda_1 e(t) = 0, \\ \ddot{e}(t) + (\lambda_1 + \lambda_2)\dot{e}(t) + \lambda_1\lambda_2 e(t) = 0, \\ \dddot{e}(t) + (\lambda_1 + \lambda_2 + \lambda_3)\ddot{e}(t) + (\lambda_1\lambda_2 + \\ \lambda_1\lambda_3 + \lambda_2\lambda_3)\dot{e}(t) + \lambda_1\lambda_2\lambda_3 e(t) = 0, \\ e^{(n)}(t) + \sum_{i=1}^{n-1} e^{(i)}(t)\big(\sum_{j=1}^{n}\sum_{z=j+1}^{n}\cdots \\ \sum_{w=n-i}^{n}\lambda_j\lambda_z\ldots\lambda_w\big) + \lambda_1\ldots\lambda_n e(t) = 0. \end{cases}$$

### 3.2 Standard ZE Inequations of UPV Situation

In this sector, we specially examine the formulas for the 0-order, 1-order, 2-order, 3-order, and $n$-order about inequations.

1) Order-0 ZE General Formula of UPV Situation:

$$e(t) \le 0. \tag{20}$$

Based on ZE, the given formula is applicable as $t \gg 0$.

2) Order-1 ZE General Formula of UPV Situation: According to ZE, we have the following formulas:

$$\dot{e}(t) + \lambda_1 e(t) \le 0, \tag{21}$$

$$\dot{a}(t) - \dot{d}(t) + \lambda_1(a(t) - d(t)) \le 0. \tag{22}$$

3) Order-2 ZE General Formula of UPV Situation:

$$\ddot{e}(t) + (\lambda_1 + \lambda_2)\dot{e}(t) + \lambda_1\lambda_2 e(t) \le 0. \tag{23}$$

Similarly, we set $\epsilon_1(t)$ equal to $\dot{e}(t) + \lambda_1 e(t)$ and incorporate this into $\dot{\epsilon}_1(t) + \lambda_1\epsilon_1(t) \le 0$. Through the process of simplification, we obtain (23).

4) 3-order ZE General Formula of UPV Situation:

$$\dddot{e}(t) + (\lambda_1 + \lambda_2 + \lambda_3)\ddot{e}(t) + (\lambda_1\lambda_2 + \\ \lambda_1\lambda_3 + \lambda_2\lambda_3)\dot{e}(t) + \lambda_1\lambda_2\lambda_3 e(t) \le 0. \tag{24}$$

By the same rationale, we set $\epsilon_2(t)$ to $\ddot{e}(t) + (\lambda_1 + \lambda_2)\dot{e}(t) + \lambda_1\lambda_2 e(t)$ and substitute this into $\dot{\epsilon}_2(t) + \lambda_3\epsilon_2(t) \le 0$, which allows us to obtain (26).

5) Order-$n$ ZE General Formula of UPV Situation:

$$e^{(n)}(t) + \sum_{i=1}^{n-1} e^{(i)}(t)\bigg(\sum_{j=1}^{n}\sum_{z=j+1}^{n}\cdots\sum_{w=n-i}^{n}\lambda_j\lambda_z$$
$$\cdots\lambda_w\bigg) + \lambda_1\ldots\lambda_n e(t) \le 0. \tag{25}$$

We define (25) as $\epsilon_n(t)$. Then we utilize mathematical induction to establish the following inequation.

$$\dot{\epsilon}_n(t) + \lambda_{n+1}\epsilon_n(t) \le 0.$$

$$e^{(n+1)}(t) + \sum_{i=1}^{n-1} e^{(i+1)}(t)\bigg(\sum_{j=1}^{n}\sum_{z=j+1}^{n}\cdots\sum_{w=n-i}^{n}\lambda_j\lambda_z$$
$$\cdots\lambda_w\bigg) + \lambda_1\ldots\lambda_n e^{(1)}(t) + \lambda_{n+1}\bigg(e^{(n)}(t) +$$
$$\sum_{i=1}^{n-1} e^{(i)}(t)\bigg(\sum_{j=1}^{n}\sum_{z=j+1}^{n}\cdots\sum_{w=n-i}^{n}\lambda_j\lambda_z\ldots\lambda_w\bigg)$$
$$+\lambda_1\ldots\lambda_n e(t)\bigg) \le 0. \tag{26}$$

Through a straightforward simplification, we arrive at the following formula.

$$e^{(n+1)}(t) + \sum_{i=1}^{n} e^{(i)}(t)\bigg(\sum_{j=1}^{n+1}\sum_{z=j+1}^{n+1}\cdots\sum_{w=n+1-i}^{n+1}\lambda_j\lambda_z$$
$$\cdots\lambda_w\bigg) + \lambda_1\ldots\lambda_{n+1}e(t) \le 0.$$

Therefore, we can conclude that the formula for the $n+1$ order is valid. When $\lambda_1 \gg 0$, $\lambda_2 \gg 0$, $\ldots, \lambda_n \gg 0$ and $t \to \infty$, the derived inequations are equivalent.

$$
\begin{cases}
e(t) \leq 0, \\
\dot{e}(t) + \lambda_1 e(t) \leq 0, \\
\ddot{e}(t) + (\lambda_1 + \lambda_2)\dot{e}(t) + \lambda_1\lambda_2 e(t) \leq 0, \\
\dddot{e}(t) + (\lambda_1 + \lambda_2 + \lambda_3)\ddot{e}(t) + (\lambda_1\lambda_2 + \\
\lambda_1\lambda_3 + \lambda_2\lambda_3)\dot{e}(t) + \lambda_1\lambda_2\lambda_3 e(t) \leq 0, \\
e^{(n)}(t) + \sum_{i=1}^{n-1} e^{(i)}(t)\left(\sum_{j=1}^{n}\sum_{z=j+1}^{n} \\
\ldots \sum_{w=n-i}^{n} \lambda_j\lambda_z \ldots \lambda_w\right) + \lambda_1 \ldots \lambda_n e(t) \leq 0.
\end{cases}
$$

## 4 Once-Integral ZE Equations of UPV Situation

In this sector, once-integral ZE equations of UPV situation are discussed. Besides, we specially examine the formulas for the -1-order, 0-order, 1-order, 2-order, and $n$-order.

1) Order-(-1) Once-Intergel General Formula: By virtue of ZE, the function $e(t)$ can be defined with $\int_0^t e(\tau)\mathrm{d}\tau$. Besides, the formula also can be definde with $\int_0^t a(\tau) - d(\tau)\mathrm{d}\tau$. The subsequent mathematical expression is obtained, when $t \to \infty$.

$$
\int_0^t e(\tau)\mathrm{d}\tau = 0. \tag{27}
$$

2) Order-0 Once-Integral General Formula:

$$
e(t) + \lambda_1 \int_0^t e(\tau)\mathrm{d}\tau = 0. \tag{28}
$$

By defining $\epsilon_1(t)$ in $\dot{\epsilon}_1(t) + \lambda_1\epsilon_1(t) = 0$ as $\int_0^t e(\tau)\mathrm{d}\tau$, we arrive at (28) as $\lambda_1 \gg 0$, $\lambda_2 \gg 0$, $\lambda_3 \gg 0$ and $t \to \infty$.

3) Order-1 Once-Integral General Formula:

$$
\dot{e}(t) + (\lambda_1 + \lambda_2)e(t) + \lambda_1\lambda_2 \int_0^t e(\tau)\mathrm{d}\tau = 0. \tag{29}
$$

In the case of the order-1 formulation, we consider $\int_0^t \epsilon_2(\tau)\mathrm{d}\tau = e(t) + \lambda_1 \int_0^t e(\tau)\mathrm{d}\tau$ and subsequently incorporate $\int_0^t \epsilon_2(\tau)\mathrm{d}\tau$ into $\epsilon_2(t) + \lambda_2 \int_0^t \epsilon_2(\tau)\mathrm{d}\tau = 0$, when $\lambda_1 \gg 0$, $\lambda_2 \gg 0$ and $t \to \infty$. Then, we arrive at (29).

4) Order-2 Once-Intergel General Formula:

$$
\ddot{e}(t) + (\lambda_1 + \lambda_2 + \lambda_3)\dot{e}(t) + (\lambda_1\lambda_2 + \lambda_1\lambda_3 + \lambda_2\lambda_3)
$$
$$
e(t) + \lambda_1\lambda_2\lambda_3 \int_0^t e(\tau)\mathrm{d}\tau = 0. \tag{30}
$$

We propose setting $\int_0^t \epsilon_3(\tau)\mathrm{d}\tau$ to be equal to $\dot{e}(t) + (\lambda_1 + \lambda_2)e(t) + \lambda_1\lambda_2 \int_0^t e(\tau)\mathrm{d}\tau$, and this expression is then substituted into formula $\epsilon_3(t) + \lambda_3 \int_0^t \epsilon_3(\tau)\mathrm{d}\tau = 0$. Then we arrive at (30).

5) Order-$n$ Once-Integral General Formula:

$$
e^{(n)}(t) + \sum_{i=1}^{n}\left(\int_0^t e(\tau)\mathrm{d}\tau\right)^{(i)}\left(\sum_{j=1}^{n+1}\sum_{z=j+1}^{n+1}\cdots\right.
$$
$$
\left.\sum_{w=n+1-i}^{n+1} \lambda_j\lambda_z \ldots \lambda_w\right) + \lambda_1 \ldots \lambda_{n+1} \int_0^t e(\tau)\mathrm{d}\tau = 0. \tag{31}
$$

We adopt the technique of mathematical induction. We start by assuming the correctness of the formula for the $(n-1)$th order. We subsequently set $\int_0^t \epsilon_n(\tau)\mathrm{d}\tau$ to be equal to

$$
e^{(n-1)}(t) + \sum_{i=1}^{n-1}\left(\int_0^t e(\tau)\mathrm{d}\tau\right)^{(i)}\left(\sum_{j=1}^{n}\sum_{z=j+1}^{n}\cdots\right.
$$
$$
\left.\sum_{w=n-i}^{n} \lambda_j\lambda_z \ldots \lambda_w\right) + \lambda_1 \ldots \lambda_n \int_0^t e(\tau)\mathrm{d}\tau
$$

and integrate this assignment into $\epsilon_n(t) + \lambda_{n+1} \int_0^t \epsilon_n(\tau)\mathrm{d}\tau = 0$, which leads to the derivation of the formula for the $n$ order.

When $\lambda_1 \gg 0$, $\lambda_2 \gg 0$, $\ldots \lambda_{n+1} \gg 0$ and $t \to \infty$, the derived equations are equivalent.

$$
\begin{cases}
\int_0^t e(\tau)\mathrm{d}\tau = 0, \\
e(t) + \lambda_1 \int_0^t e(\tau)\mathrm{d}\tau = 0, \\
\dot{e}(t) + (\lambda_1 + \lambda_2)e(t) + \lambda_1\lambda_2 \int_0^t e(\tau)\mathrm{d}\tau = 0, \\
\ddot{e}(t) + (\lambda_1 + \lambda_2 + \lambda_3)\dot{e}(t) + (\lambda_1\lambda_2 + \lambda_1\lambda_3 + \\
\lambda_2\lambda_3)e(t) + \lambda_1\lambda_2\lambda_3 \int_0^t e(\tau)\mathrm{d}\tau = 0, \\
e^{(n)}(t) + \sum_{i=1}^{n}(\int_0^t e(\tau)\mathrm{d}\tau)^{(i)}\left(\sum_{j=1}^{n+1}\sum_{z=j+1}^{n+1}\cdots\right. \\
\left.\sum_{w=n+1-i}^{n+1} \lambda_j\lambda_z \ldots \lambda_w\right) + \lambda_1 \ldots \lambda_{n+1} \int_0^t e(\tau)\mathrm{d}\tau \\
= 0.
\end{cases}
$$

## 5 Once-Integral ZE Inequations of UPV Situation

In this sector, once-integral ZE inequations of UPV situation are discussed. Besides, we specially examine the formulas for the order(-1), order-0, order-1, order-2, and order-$n$.

1) Order-(-1) Once-Integral General Formula:
In the context of $t \gg 0$, we define $e(t)$ in $e(t) \leq 0$ as $\int_0^t e(\tau)\mathrm{d}\tau$.

$$
\int_0^t e(\tau)\mathrm{d}\tau \leq 0. \tag{32}
$$

2) Order-0 Once-Integral General Formula:

$$
e(t) + \lambda_1 \int_0^t e(\tau)\mathrm{d}\tau \leq 0. \tag{33}
$$

By defining $\epsilon_1(t)$ in $\dot{\epsilon}_1(t) + \lambda_1\epsilon_1(t) \leq 0$ as $\int_0^t e(\tau)\mathrm{d}\tau$, we arrive at (33).

3) Order-1 Once-Integral General Formula:

$$
\dot{e}(t) + (\lambda_1 + \lambda_2)e(t) + \lambda_1\lambda_2 \int_0^t e(\tau)\mathrm{d}\tau \leq 0. \tag{34}
$$

We consider $\int_0^t \epsilon_2(\tau)\mathrm{d}\tau = e(t) + \lambda_1 \int_0^t e(\tau)\mathrm{d}\tau$ and subsequently incorporate it into $\epsilon_2(t) + \lambda_2 \int_0^t \epsilon_2(\tau)\mathrm{d}\tau \leq 0$, as $t \to \infty$ and $\lambda_1 \gg 0$.

4) Order-2 Once-Integral General Formula:

$$\ddot{e}(t) + (\lambda_1 + \lambda_2 + \lambda_3)\dot{e}(t) + (\lambda_1\lambda_2 + \lambda_1\lambda_3 + \lambda_2\lambda_3)$$
$$e(t) + \lambda_1\lambda_2\lambda_3 \int_0^t e(\tau)\mathrm{d}\tau \leq 0. \tag{35}$$

The derivation process is in line with the methodology employed in the preceding derivations. It can be established that when $\lambda_1 \gg 0$, $\lambda_2 \gg 0$, $\lambda_3 \gg 0$ and $t \to \infty$, there exists an equivalence among (32), (33), (34), and (35).

5) Order-$n$ Once-Integral General Formula:

$$e^{(n)}(t) + \sum_{i=1}^{n} \left( \int_0^t e(\tau)\mathrm{d}\tau \right)^{(i)} \left( \sum_{j=1}^{n+1} \sum_{z=j+1}^{n+1} \cdots \right.$$
$$\left. \sum_{w=n+1-i}^{n+1} \lambda_j\lambda_z\ldots\lambda_w \right) + \lambda_1\ldots\lambda_{n+1} \int_0^t e(\tau)\mathrm{d}\tau \leq 0. \tag{36}$$

We adopt the technique of mathematical induction. We start by assuming the correctness of the formula for the $(n-1)$th order. We subsequently set $\int_0^t \epsilon_{n-1}(\tau)$ to be equal to

$$e^{(n-1)}(t) + \sum_{i=1}^{n-1} \left( \int_0^t e(\tau)\mathrm{d}\tau \right)^{(i)} \left( \sum_{j=1}^{n} \sum_{z=j+1}^{n} \cdots \right.$$
$$\left. \sum_{w=n-i}^{n} \lambda_j\lambda_z\ldots\lambda_w \right) + \lambda_1\ldots\lambda_n \int_0^t e(\tau)\mathrm{d}\tau$$

and integrate this assignment into $\epsilon_{n-1}(t) + \lambda_{n+1} \int_0^t \epsilon_{n-1}(\tau)\mathrm{d}\tau \leq 0$, which leads to the derivation of the formula for the $n$th order.

When $\lambda_1 \gg 0$, $\lambda_2 \gg 0$, ..., $\lambda_{n+1} \gg 0$ and $t \to \infty$, the derived inequations are euqivalent.

$$\begin{cases} \int_0^t e(\tau)\mathrm{d}\tau \leq 0, \\ e(t) + \lambda_1 \int_0^t e(\tau)\mathrm{d}\tau \leq 0, \\ \dot{e}(t) + (\lambda_1 + \lambda_2)e(t) + \lambda_1\lambda_2 \int_0^t e(\tau)\mathrm{d}\tau \leq 0, \\ \ddot{e}(t) + (\lambda_1 + \lambda_2 + \lambda_3)\dot{e}(t) + (\lambda_1\lambda_2 + \lambda_1\lambda_3 + \\ \lambda_2\lambda_3)e(t) + \lambda_1\lambda_2\lambda_3 \int_0^t e(\tau)\mathrm{d}\tau \leq 0, \\ e^{(n)}(t) + \sum_{i=1}^{n}(\int_0^t e(\tau)\mathrm{d}\tau)^{(i)} \left( \sum_{j=1}^{n+1} \sum_{z=j+1}^{n+1} \cdots \right. \\ \left. \sum_{w=n+1-i}^{n+1} \lambda_j\lambda_z\ldots\lambda_w \right) + \lambda_1\ldots\lambda_{n+1} \int_0^t e(\tau)\mathrm{d}\tau \\ \leq 0. \end{cases}$$

## 6 Twice-Integral ZE Equations of UPV Situation

In this section, we discuss twice-integral ZE equations of UPV. We also derive and analyze the formulas pertaining to twice-integral of various orders, specifically -2, -1, 0, and $n$.

1) Order-(-2) Twice-Integral General Formula:

$$\int_0^t \mathrm{d}\tau_1 \int_0^{\tau_1} e(\tau_0)\mathrm{d}\tau_0 = 0. \tag{37}$$

In the context of $t \gg 0$, we define $\epsilon_0(t)$ in $\epsilon_0(t) = 0$ as $\int_0^t \mathrm{d}\tau_1 \int_0^{\tau_1} e(\tau_0)\mathrm{d}\tau_0$.

2) Order-(-1) Twice-Integral General Formula:

$$\int_0^t e(\tau_1)\mathrm{d}\tau_1 + \lambda_1 \int_0^t \mathrm{d}\tau_1 \int_0^{\tau_1} e(\tau_0)\mathrm{d}\tau_0 = 0. \tag{38}$$

By defining $\epsilon_1(t)$ in $\dot{\epsilon}_1(t) + \lambda_1\epsilon_1(t) = 0$ as $\int_0^t \mathrm{d}\tau_1 \int_0^{\tau_1} e(\tau_0)\mathrm{d}\tau_0$, we arrive at (38).

3) Order-0 Twice-Integral General Formula:

By defining $\epsilon_2(t)$ in $\dot{\epsilon}_2(t) + \lambda_2\epsilon_2(t) = 0$ as $\int_0^t e(\tau_1)\mathrm{d}\tau_1 + \lambda_1 \int_0^t \mathrm{d}\tau_1 \int_0^{\tau_1} e(\tau_0)\mathrm{d}\tau_0$, we arrive at the following expression when $\lambda_1 \gg 0$, $\lambda_2 \gg 0$ and $t \to \infty$:

$$e(t) + (\lambda_1 + \lambda_2) \int_0^t e(\tau_1)d(\tau_1) + \lambda_1\lambda_2$$
$$\int_0^t \mathrm{d}\tau_1 \int_0^{\tau_1} e(\tau_0)\mathrm{d}\tau_0 = 0. \tag{39}$$

4) Order-1 Twice-Integral General Formula:

By defining $\epsilon_3(t)$ in $\dot{\epsilon}_3(t) + \lambda_3\epsilon_3(t) = 0$ as $e(t) + (\lambda_1+\lambda_2) \int_0^t e(\tau_1)d(\tau_1) + \lambda_1\lambda_2 \int_0^t \mathrm{d}\tau_1 \int_0^{\tau_1} e(\tau_0)\mathrm{d}\tau_0$, we arrive at the following formula when $\lambda_1 \gg 0$, $\lambda_2 \gg 0$, $\lambda_3 \gg 0$ and $t \to \infty$:

$$\dot{e}(t) + (\lambda_1 + \lambda_2 + \lambda_3)e(t) + (\lambda_1\lambda_2 + \lambda_1\lambda_3 + \lambda_2\lambda_3)$$
$$\int_0^t e(\tau_1)d\tau_1 + \lambda_1\lambda_2\lambda_3 \int_0^t \mathrm{d}\tau_1 \int_0^{\tau_1} e(\tau_0)\mathrm{d}\tau_0 = 0. \tag{40}$$

5) Order-$n$ Twice-Integral General Formula:

$$e^{(n)}(t) + \sum_{i=1}^{n+1} \left( \int_0^t \mathrm{d}\tau_1 \int_0^{\tau_1} e(\tau_0)d\tau_0 \right)^{(i)} \sum_{j=1}^{n+2} \sum_{z=j+1}^{n+2}$$
$$\cdots \sum_{w=n+2-i}^{n+2} \lambda_j\lambda_z\ldots\lambda_w + \lambda_1\lambda_2\ldots\lambda_{n+2}$$
$$\left( \int_0^t \mathrm{d}\tau_1 \int_0^{\tau_1} e(\tau_0)\mathrm{d}\tau_0 \right) = 0. \tag{41}$$

We also use mathematical induction to prove the assertion. We suppose that the formula is valid for the $(n-1)$th order. By incorporating this into formula $\dot{e}(t) + \lambda_n e(t) = 0$, we can obtain (41) as a consequence.

When $\lambda_1 \gg 0$, $\lambda_2 \gg 0$, ..., $\lambda_{n+2} \gg 0$ and $t \to \infty$, the derived equations are equivalent.

$$\begin{cases} \int_0^t \mathrm{d}\tau_1 \int_0^{\tau_1} e(\tau_0)\mathrm{d}\tau_0 = 0, \\ \int_0^t e(\tau_1)\mathrm{d}\tau_1 + \lambda_1 \int_0^t \mathrm{d}\tau_1 \int_0^{\tau_1} e(\tau_0)\mathrm{d}\tau_0 = 0, \\ e(t) + (\lambda_1 + \lambda_2) \int_0^t e(\tau_1)d(\tau_1) + \\ \lambda_1\lambda_2 \int_0^t \mathrm{d}\tau_1 \int_0^{\tau_1} e(\tau_0)\mathrm{d}\tau_0 = 0, \\ \dot{e}(t) + (\lambda_1 + \lambda_2 + \lambda_3)e(t) + \\ (\lambda_1\lambda_2 + \lambda_1\lambda_3 + \lambda_2\lambda_3) \int_0^t e(\tau_1)d\tau_1 + \\ \lambda_1\lambda_2\lambda_3 \int_0^t \mathrm{d}\tau_1 \int_0^{\tau_1} e(\tau_0)\mathrm{d}\tau_0 = 0, \\ e^{(n)}(t) + \sum_{i=1}^{n+1}(\int_0^t \mathrm{d}\tau_1 \int_0^{\tau_1} e(\tau_0)\mathrm{d}\tau_0)^{(i)} \sum_{j=1}^{n+2} \\ \sum_{z=j+1}^{n+2} \cdots \sum_{w=n+2-i}^{n+2} \lambda_j\lambda_z\ldots\lambda_w + \lambda_1\ldots\lambda_{n+2} \\ (\int_0^t \mathrm{d}\tau_1 \int_0^{\tau_1} e(\tau_0)\mathrm{d}\tau_0) = 0. \end{cases}$$

# 7 Twice-Integral ZE Inequations of UPV Situation

In this section, we discuss twice-integral ZE inequations of UPV. We also derive and analyze the formulas pertaining to twice-integral of various orders, specifically -2, -1, 0, and $n$.

1) Order-(-2) Twice-Integral General Formula:

$$\int_0^t d\tau_1 \int_0^{\tau_1} e(\tau_0)d\tau_0 \leq 0. \qquad (42)$$

According to ZE, when the condition of $t \to \infty$, (44) is valid.

2) Order-(-1) Twice-Integral General Formula:
By defining $\epsilon_1(t)$ in $\dot{\epsilon}_1(t) + \lambda_1\epsilon_1(t) \leq 0$ as $\int_0^t d\tau_1 \int_0^{\tau_1} e(\tau_0)d\tau_0$, we arrive at the following expression as $\lambda_1 \gg 0$ and $t \to \infty$ and:

$$\int_0^t e(\tau_1)d\tau_1 + \lambda_1 \int_0^t d\tau_1 \int_0^{\tau_1} e(\tau_0)d\tau_0 \leq 0. \quad (43)$$

3) Order-0 Twice-Integral General Formula:
By defining $\epsilon_2(t)$ in $\dot{\epsilon}_2(t) + \lambda_2\epsilon_2(t)leq0$ as $\int_0^t e(\tau_1)d\tau_1 + \lambda_1 \int_0^t d\tau_1 \int_0^{\tau_1} e(\tau_0)d\tau_0$, we arrive at the following expression as $\lambda_1 \gg 0$, $\lambda_2 \gg 0$ and $t \to \infty$:

$$e(t) + (\lambda_1 + \lambda_2)\int_0^t e(\tau_1)d(\tau_1) + \lambda_1\lambda_2 \\ \int_0^t d\tau_1 \int_0^{\tau_1} e(\tau_0)d\tau_0 \leq 0. \qquad (44)$$

4) Order-1 Twice-Integral General Formula:
By defining $\epsilon_3(t)$ in $\dot{\epsilon}_3(t) + \lambda_3\epsilon_3(t) \leq 0$ as $e(t) + (\lambda_1+\lambda_2)\int_0^t e(\tau_1)d(\tau_1) + \lambda_1\lambda_2\int_0^t d\tau_1 \int_0^{\tau_1} e(\tau_0)d\tau_0$, we arrive at the following expression as $\lambda_1 \gg 0$ $\lambda_2 \gg 0$, $\lambda_3 \gg 0$ and $t \to \infty$:

$$\dot{e}(t) + (\lambda_1 + \lambda_2 + \lambda_3)e(t) + (\lambda_1\lambda_2 + \lambda_1\lambda_3 + \lambda_2\lambda_3) \\ \int_0^t e(\tau_1)d\tau_1 + \lambda_1\lambda_2\lambda_3 \int_0^t d\tau_1 \int_0^{\tau_1} e(\tau_0)d\tau_0 \leq 0. \qquad (45)$$

5) Order-$n$ Twice-Integral General Formula:

$$e^{(n)}(t) + \sum_{i=1}^{n+1}\Big(\int_0^t d\tau_1 \int_0^{\tau_1} e(\tau_0)d\tau_0\Big)^{(i)} \sum_{j=1}^{n+2}\sum_{z=j+1}^{n+2} \\ \dots \sum_{w=n+2-i}^{n+2} \lambda_j\lambda_z\dots\lambda_w + \lambda_1\lambda_2\dots\lambda_{n+2} \\ \Big(\int_0^t d\tau_1 \int_0^{\tau_1} e(\tau_0)d\tau_0\Big) \leq 0. \qquad (46)$$

We suppose that the formula is valid for the $n-1$th order. By incorporating this into formula $\dot{\epsilon}_{n+2}t) + \lambda_{n+2}\epsilon_{n+2}(t) \leq 0$, we can obtain (46).
When $\lambda_1 \gg 0$, $\lambda_2 \gg 0$, ..., $\lambda_{n+2} \gg 0$ and $t \to \infty$, the derived inequations are equivalent.

$$\begin{cases} \int_0^t d\tau_1 \int_0^{\tau_1} e(\tau_0)d\tau_0 \leq 0, \\ \int_0^t e(\tau_1)d\tau_1 + \lambda_1 \int_0^t d\tau_1 \int_0^{\tau_1} e(\tau_0)d\tau_0 \leq 0, \\ e(t) + (\lambda_1 + \lambda_2)\int_0^t e(\tau_1)d(\tau_1) + \\ \lambda_1\lambda_2 \int_0^t d\tau_1 \int_0^{\tau_1} e(\tau_0)d\tau_0 \leq 0, \\ \dot{e}(t) + (\lambda_1 + \lambda_2 + \lambda_3)e(t) + \\ (\lambda_1\lambda_2 + \lambda_1\lambda_3 + \lambda_2\lambda_3)\int_0^t e(\tau_1)d\tau_1 + \\ \lambda_1\lambda_2\lambda_3 \int_0^t d\tau_1 \int_0^{\tau_1} e(\tau_0)d\tau_0 \leq 0, \\ e^{(n)}(t) + \sum_{i=1}^{n+1}(\int_0^t d\tau_1 \int_0^{\tau_1} e(\tau_0)d\tau_0)^{(i)} \sum_{j=1}^{n+2} \\ \sum_{z=j+1}^{n+2}\dots\sum_{w=n+2-i}^{n+2} \lambda_j\lambda_z\dots\lambda_w + \lambda_1\dots\lambda_{n+2} \\ (\int_0^t d\tau_1 \int_0^{\tau_1} e(\tau_0)d\tau_0) \leq 0. \end{cases}$$

# 8 Thrice-Integral ZE Equations of UPV Situation

In this section, we discuss thrice-integral ZE equations of UPV. We also derive and analyze the formulas pertaining to thrice-integral of various orders, specifically -3, -2, -1, and $n$.

1) Order-(-3) Thrice-Integral General Formula:

$$\int_0^t d\tau_2 \int_0^{\tau_2} d\tau_1 \int_0^{\tau_1} e(\tau_0)d\tau_0 = 0. \qquad (47)$$

In the context of $t \gg 0$, we define $\epsilon_0(t)$ in $\epsilon_0(t) = 0$ as $\int_0^t d\tau_2 \int_0^{\tau_2} d\tau_1 \int_0^{\tau_1} e(\tau_0)d\tau_0$.

2) Order-(-2) Thrice-Integral General Formula:
By defining $e(t)$ in $\dot{\epsilon}_1(t) + \lambda_1\epsilon_1(t) = 0$ as $\int_0^t d\tau_2 \int_0^{\tau_2} d\tau_1 \int_0^{\tau_1} e(\tau_0)d\tau_0$, we arrive at the following expression as $\lambda_1 \gg 0$ and $t \to \infty$:

$$\int_0^t d\tau_2 \int_0^{\tau_2} e(\tau_1)d\tau_1 + \lambda_1 \int_0^t d\tau_2 \int_0^{\tau_2} d\tau_1 \\ \int_0^{\tau_1} e(\tau_0)d\tau_0 = 0. \qquad (48)$$

3) Order-(-1) Thrice-Integral General Formula:
By defining $e(t)$ in $\dot{\epsilon}_2(t) + \lambda_2\epsilon_2(t) = 0$ as $\int_0^t d\tau_2 \int_0^{\tau_2} e(\tau_1)d\tau_1 + \lambda_1 \int_0^t d\tau_2 \int_0^{\tau_2} d\tau_1 \int_0^{\tau_1} e(\tau_0)d\tau_0$, we arrive at the following formula as $\lambda_1 \gg 0$, $\lambda_2 \gg 0$ and $t \to \infty$:

$$\int_0^t e(\tau_2)d\tau_2 + (\lambda_1 + \lambda_2)\int_0^t d\tau_2 \int_0^{\tau_2} e(\tau_1)d(\tau_1) + \\ \lambda_1\lambda_2 \int_0^t d\tau_2 \int_0^{\tau_2} d\tau_1 \int_0^{\tau_1} e(\tau_0)d\tau_0 = 0. \qquad (49)$$

4) Order-0 Thrice-Integral General Formula:
According to the previous method, we arrive at the following expression as $\lambda_1 \gg 0$, $\lambda_2 \gg 0$, $\lambda_3 \gg 0$ and $t \to \infty$:

$$e(t) + (\lambda_1 + \lambda_2 + \lambda_3)\int_0^t e(\tau_2)d\tau_1 + (\lambda_1\lambda_2 + \\ \lambda_1\lambda_3 + \lambda_2\lambda_3)\int_0^t d\tau_2 \int_0^{\tau_2} e(\tau_1)d\tau_1 + \lambda_1\lambda_2\lambda_3 \\ \int_0^t d\tau_2 \int_0^{\tau_2} d\tau_1 \int_0^{\tau_1} e(\tau_0)d\tau_0 = 0. \qquad (50)$$

5) Order-$n$ Thrice-Integral General Formula:

$$e^{(n)}(t) + \sum_{i=1}^{n+2} \left( \int_0^t \mathrm{d}\tau_2 \int_0^{\tau_2} \mathrm{d}\tau_1 \int_0^{\tau_1} e(\tau_0) d\tau_0 \right)^{(i)}$$

$$\sum_{j=1}^{n+3} \sum_{z=j+1}^{n+3} \cdots \sum_{w=n+3-i}^{n+3} \lambda_j \lambda_z \ldots \lambda_w + \lambda_1 \ldots \lambda_{n+3}$$

$$\left( \int_0^t \mathrm{d}\tau_2 \int_0^{\tau_2} \mathrm{d}\tau_1 \int_0^{\tau_1} e(\tau_0) \mathrm{d}\tau_0 \right) = 0. \tag{51}$$

We also use mathematical induction to prove the assertion. We suppose that the formula is valid for the $(n-1)$th order. By incorporating this into formula $\dot{\epsilon}_{n+3}(t) + \lambda_{n+3}\epsilon_{n+3}(t) = 0$, we can obtain (41) as a consequence.

Additionally, when $\lambda_1 \gg 0$, $\lambda_2 \gg 0$, ..., $\lambda_{n+3} \gg 0$ and $t \to \infty$ , the derived equations are equivalent.

$$
\begin{cases}
\int_0^t \mathrm{d}\tau_2 \int_0^{\tau_2} \mathrm{d}\tau_1 \int_0^{\tau_1} e(\tau_0)\mathrm{d}\tau_0 = 0, \\[4pt]
\int_0^t \mathrm{d}\tau_2 \int_0^{\tau_2} e(\tau_1)\mathrm{d}\tau_1 + \\
\lambda_1 \int_0^t \mathrm{d}\tau_2 \int_0^{\tau_2} \mathrm{d}\tau_1 \int_0^{\tau_1} e(\tau_0)\mathrm{d}\tau_0 = 0, \\[4pt]
\int_0^t e(\tau_2)\mathrm{d}\tau_2 + (\lambda_1 + \lambda_2) \int_0^t \mathrm{d}\tau_2 \int_0^{\tau_2} e(\tau_1)d(\tau_1) + \\
\lambda_1 \lambda_2 \int_0^t \mathrm{d}\tau_2 \int_0^{\tau_2} \mathrm{d}\tau_1 \int_0^{\tau_1} e(\tau_0)\mathrm{d}\tau_0 = 0, \\[4pt]
e(t) + (\lambda_1 + \lambda_2 + \lambda_3) \int_0^t e(\tau_2)\mathrm{d}\tau_1 + (\lambda_1\lambda_2 + \\
\lambda_1\lambda_3 + \lambda_2\lambda_3) \int_0^t \mathrm{d}\tau_2 \int_0^{\tau_2} e(\tau_1)d\tau_1 + \lambda_1\lambda_2\lambda_3 \\
\int_0^t \mathrm{d}\tau_2 \int_0^{\tau_2} \mathrm{d}\tau_1 \int_0^{\tau_1} e(\tau_0)\mathrm{d}\tau_0 = 0, \\[4pt]
e^{(n)}(t) + \sum_{i=1}^{n+2}(\int_0^t \mathrm{d}\tau_2 \int_0^{\tau_2} \mathrm{d}\tau_1 \int_0^{\tau_1} e(\tau_0)d\tau_0)^{(i)} \\
\sum_{j=1}^{n+3} \sum_{z=j+1}^{n+3} \cdots \sum_{w=n+3-i}^{n+3} \lambda_j \lambda_z \ldots \lambda_w + \\
\lambda_1 \ldots \lambda_{n+3}(\int_0^t \mathrm{d}\tau_2 \int_0^{\tau_2} \mathrm{d}\tau_1 \int_0^{\tau_1} e(\tau_0)\mathrm{d}\tau_0) = 0.
\end{cases}
$$

## 9  Thrice-Integral ZE Inequations of UPV Situation

In this section, we discuss thrice-integral ZE inequations of UPV. We also derive and analyze the formulas pertaining to thrice-integral of various orders, specifically -3, -2, -1, and $n$.

1) Order-(-3) Thrice-Integral ZE General Formula:

$$\int_0^t \mathrm{d}\tau_2 \int_0^{\tau_2} d\tau_1 \int_0^{\tau_1} e(\tau_0)\mathrm{d}\tau_0 \le 0. \tag{52}$$

In the context of $t \gg 0$, we define $\epsilon_0(t)$ in $\epsilon_0(t) \le 0$ as $\int_0^t \mathrm{d}\tau_2 \int_0^{\tau_2} d\tau_1 \int_0^{\tau_1} e(\tau_0)\mathrm{d}\tau_0$.

2) Order-(-2) Thrice-Integral General Formula:
By defining $e(t)$ in $\dot{\epsilon}_1(t) + \lambda_1\epsilon_1(t) \le 0$ as $\int_0^t \mathrm{d}\tau_2 \int_0^{\tau_2} d\tau_1 \int_0^{\tau_1} e(\tau_0)\mathrm{d}\tau_0$, we arrive at the following expression as $\lambda_1 \gg 0$ and $t \to \infty$:

$$\int_0^t \mathrm{d}\tau_2 \int_0^{\tau_2} e(\tau_1)\mathrm{d}\tau_1 + \lambda_1 \int_0^t \mathrm{d}\tau_2 \int_0^{\tau_2} d\tau_1$$

$$\int_0^{\tau_1} e(\tau_0)\mathrm{d}\tau_0 \le 0. \tag{53}$$

3) Order-(-1) Thrice-Integral General Formula:
By defining $e(t)$ in $\dot{\epsilon}_2(t) + \lambda_2\epsilon_2(t) \le 0$ as $\int_0^t \mathrm{d}\tau_2 \int_0^{\tau_2} e(\tau_1)\mathrm{d}\tau_1 + \lambda_1 \int_0^t \mathrm{d}\tau_2 \int_0^{\tau_2} d\tau_1 \int_0^{\tau_1} e(\tau_0)\mathrm{d}\tau_0$,

we arrive at the following formula as $\lambda_1 \gg 0$, $\lambda_2 \gg 0$ and $t \to \infty$:

$$\int_0^t e(\tau_2)\mathrm{d}\tau_2 + (\lambda_1 + \lambda_2) \int_0^t \mathrm{d}\tau_2 \int_0^{\tau_2} e(\tau_1)d(\tau_1) +$$

$$\lambda_1\lambda_2 \int_0^t \mathrm{d}\tau_2 \int_0^{\tau_2} \mathrm{d}\tau_1 \int_0^{\tau_1} e(\tau_0)\mathrm{d}\tau_0 \le 0. \tag{54}$$

4) Order-0 Thrice-Integral General Formula:
According to the previous method, we arrive at the following expression as $\lambda_1 \gg 0$, $\lambda_2 \gg 0$, $\lambda_3 \gg 0$ and $t \to \infty$:

$$e(t) + (\lambda_1 + \lambda_2 + \lambda_3) \int_0^t e(\tau_2)\mathrm{d}\tau_1 + (\lambda_1\lambda_2 +$$

$$\lambda_1\lambda_3 + \lambda_2\lambda_3) \int_0^t \mathrm{d}\tau_2 \int_0^{\tau_2} e(\tau_1)d\tau_1 + \lambda_1\lambda_2\lambda_3 \tag{55}$$

$$\int_0^t \mathrm{d}\tau_2 \int_0^{\tau_2} \mathrm{d}\tau_1 \int_0^{\tau_1} e(\tau_0)\mathrm{d}\tau_0 \le 0.$$

5) Order-$n$ Thrice-Integral General Formula:

$$e^{(n)}(t) + \sum_{i=1}^{n+2} \left( \int_0^t \mathrm{d}\tau_2 \int_0^{\tau_2} \mathrm{d}\tau_1 \int_0^{\tau_1} e(\tau_0)d\tau_0 \right)^{(i)}$$

$$\sum_{j=1}^{n+3} \sum_{z=j+1}^{n+3} \cdots \sum_{w=n+3-i}^{n+3} \lambda_j \lambda_z \ldots \lambda_w + \lambda_1 \ldots \lambda_{n+3}$$

$$\left( \int_0^t \mathrm{d}\tau_2 \int_0^{\tau_2} \mathrm{d}\tau_1 \int_0^{\tau_1} e(\tau_0)d\tau_0 \right) \le 0. \tag{56}$$

We also use mathematical induction to prove the assertion. We suppose that the formula is valid for the $(n-1)$th order. By incorporating this into formula $\dot{\epsilon}_{n+3}(t) + \lambda_{n+3}\epsilon_{n+3}(t) = 0$, we can obtain (41) as a consequence.

Additionally, when $\lambda_1 \gg 0$, $\lambda_2 \gg 0$, ..., $\lambda_{n+3} \gg 0$ and $t \to \infty$ , the derived equations are equivalent.

$$
\begin{cases}
\int_0^t \mathrm{d}\tau_2 \int_0^{\tau_2} \mathrm{d}\tau_1 \int_0^{\tau_1} e(\tau_0)\mathrm{d}\tau_0 \le 0, \\[4pt]
\int_0^t \mathrm{d}\tau_2 \int_0^{\tau_2} e(\tau_1)\mathrm{d}\tau_1 + \\
\lambda_1 \int_0^t \mathrm{d}\tau_2 \int_0^{\tau_2} \mathrm{d}\tau_1 \int_0^{\tau_1} e(\tau_0)\mathrm{d}\tau_0 \le 0, \\[4pt]
\int_0^t e(\tau_2)\mathrm{d}\tau_2 + (\lambda_1 + \lambda_2) \int_0^t \mathrm{d}\tau_2 \int_0^{\tau_2} e(\tau_1)d(\tau_1) + \\
\lambda_1\lambda_2 \int_0^t \mathrm{d}\tau_2 \int_0^{\tau_2} \mathrm{d}\tau_1 \int_0^{\tau_1} e(\tau_0)\mathrm{d}\tau_0 \le 0, \\[4pt]
e(t) + (\lambda_1 + \lambda_2 + \lambda_3) \int_0^t e(\tau_2)\mathrm{d}\tau_1 + (\lambda_1\lambda_2 + \\
\lambda_1\lambda_3 + \lambda_2\lambda_3) \int_0^t \mathrm{d}\tau_2 \int_0^{\tau_2} e(\tau_1)d\tau_1 + \lambda_1\lambda_2\lambda_3 \\
\int_0^t \mathrm{d}\tau_2 \int_0^{\tau_2} \mathrm{d}\tau_1 \int_0^{\tau_1} e(\tau_0)\mathrm{d}\tau_0 \le 0, \\[4pt]
e^{(n)}(t) + \sum_{i=1}^{n+2}(\int_0^t \mathrm{d}\tau_2 \int_0^{\tau_2} \mathrm{d}\tau_1 \int_0^{\tau_1} e(\tau_0)d\tau_0)^{(i)} \\
\sum_{j=1}^{n+3} \sum_{z=j+1}^{n+3} \cdots \sum_{w=n+3-i}^{n+3} \lambda_j \lambda_z \ldots \lambda_w + \\
\lambda_1 \ldots \lambda_{n+3}(\int_0^t \mathrm{d}\tau_2 \int_0^{\tau_2} \mathrm{d}\tau_1 \int_0^{\tau_1} e(\tau_0)\mathrm{d}\tau_0) \le 0.
\end{cases}
$$

## 10  $m$-Times-Integral ZE Equations of UPV Situation

In this section, we discuss $m$-times-integral ZE equations of UPV. We also derive and analyze the formulas pertaining to $m$-integral of various orders, specifically $-m$, $-(m-1)$, $-(m-2)$, $-(m-3)$, 0 and $n$.

1) Order-$(-m)$ $m$-Integral General Formula:

$$\int_0^t \mathrm{d}\tau_{(m-1)} \int_0^{\tau_{(m-1)}} \mathrm{d}\tau_{(m-2)} \dots \int_0^{\tau_1} e(\tau_0)\mathrm{d}\tau_0 = 0.$$

(57)

In the context of $t \gg 0$, we define $\epsilon_0(t)$ in $\epsilon_0(t) = 0$ as $\int_0^t \mathrm{d}\tau_{(m-1)} \int_0^{\tau_{(m-1)}} \mathrm{d}\tau_{(m-2)} \dots \int_0^{\tau_1} e(\tau_0)\mathrm{d}\tau_0$.

2) Order-$(1-m)$ $m$-Integral General Formula:

$$\int_0^t \mathrm{d}\tau_{(m-2)} \int_0^{\tau_{(m-2)}} \mathrm{d}\tau_{(m-3)} \dots \int_0^{\tau_1} e(\tau_0)\mathrm{d}\tau_0 + \lambda_1$$
$$\int_0^t \mathrm{d}\tau_{(m-1)} \int_0^{\tau_{(m-1)}} \mathrm{d}\tau_{(m-2)} \dots \int_0^{\tau_1} e(\tau_0)\mathrm{d}\tau_0 = 0.$$

(58)

By defining $\epsilon_1(t)$ in $\dot{\epsilon}_1(t) + \lambda_1\epsilon_1(t) = 0$ as $\int_0^t \mathrm{d}\tau_{(m-1)} \int_0^{\tau_{(m-1)}} \mathrm{d}\tau_{(m-2)} \dots \int_0^{\tau_1} e(\tau_0)\mathrm{d}\tau_0$, we arrive at (58) when $\lambda_1 \gg 0$ and $t \to \infty$.

3) Order-$(2-m)$ $m$-Times-Integral General Formula: According to the previous method, we arrive at the following formula when $\lambda_1 \gg 0$, $\lambda_2 \gg 0$ and $t \to \infty$:

$$\int_0^t \mathrm{d}\tau_{(m-3)} \int_0^{\tau_{(m-3)}} \mathrm{d}\tau_{(m-4)} \dots \int_0^{\tau_1} e(\tau_0)\mathrm{d}\tau_0 +$$
$$(\lambda_1 + \lambda_2) \int_0^t \mathrm{d}\tau_{(m-2)} \int_0^{\tau_{(m-2)}} \mathrm{d}\tau_{(m-3)} \dots$$
$$\int_0^{\tau_1} e(\tau_0)d(\tau_0) + \lambda_1\lambda_2 \int_0^t \mathrm{d}\tau_{(m-1)} \int_0^{\tau_{(m-1)}} \mathrm{d}\tau_{(m-2)}$$
$$\dots \int_0^{\tau_1} e(\tau_0)\mathrm{d}\tau_0 = 0.$$

(59)

4) Order-$(3-m)$ $m$-Times-Integral General Formula: According to the previous method, we arrive at the following formula when $\lambda_1 \gg 0$, $\lambda_2 \gg 0$, $\lambda_3 \gg 0$ and $t \to \infty$:

$$\int_0^t \mathrm{d}\tau_{(m-4)} \int_0^{\tau_{(m-4)}} \mathrm{d}\tau_{(m-5)} \dots \int_0^{\tau_1} e(\tau_0)\mathrm{d}\tau_0 +$$
$$(\lambda_1 + \lambda_2 + \lambda_3) \int_0^t \mathrm{d}\tau_{(m-3)} \int_0^{\tau_{(m-3)}} \mathrm{d}\tau_{(m-4)} \dots$$
$$\int_0^{\tau_1} e(\tau_0)\mathrm{d}\tau_0 + (\lambda_1\lambda_2 + \lambda_1\lambda_3 + \lambda_2\lambda_3) \int_0^t \mathrm{d}\tau_{(m-2)}$$
$$\int_0^{\tau_{(m-2)}} \mathrm{d}\tau_{(m-3)} \dots \int_0^{\tau_1} e(\tau_0)\mathrm{d}\tau_0 + \lambda_1\lambda_2\lambda_3$$
$$\int_0^t \mathrm{d}\tau_{(m-1)} \int_0^{\tau_{(m-1)}} \mathrm{d}\tau_{(m-2)} \dots \int_0^{\tau_1} e(\tau_0)\mathrm{d}\tau_0 = 0.$$

(60)

5) Order-0 $m$-Times-Integral General Formula:

$$e(t) + \sum_{i=1}^{m-1} \left( \int_0^t \mathrm{d}\tau_{(m-1)} \int_0^{\tau_{(m-1)}} \mathrm{d}\tau_{(m-2)} \dots \right.$$
$$\left. \int_0^{\tau_1} e(\tau_0)d\tau_0 \right)^{(i)} \sum_{j=1}^m \sum_{z=j+1}^m \dots \sum_{w=m-i}^m \lambda_j\lambda_z \dots \lambda_w +$$
$$\lambda_1\lambda_2 \dots \lambda_m \int_0^t \int_0^{\tau_{(m-1)}} \dots \int_0^{\tau_1} e(\tau_0)\mathrm{d}\tau_0 = 0.$$

(61)

According to the previous method, we can obtain (61).

6) Order-$n$ $m$-Times-Integral General Formula:

$$e(t)^{(n)} + \sum_{i=1}^{n+m-1} \left( \int_0^t \mathrm{d}\tau_{(m-1)} \int_0^{\tau_{(m-1)}} \mathrm{d}\tau_{(m-2)} \right.$$
$$\left. \dots \int_0^{\tau_1} e(\tau_0)d\tau_0 \right)^{(i)} \sum_{j=1}^{n+m} \sum_{z=j+1}^{n+m} \dots \sum_{w=n+m-i}^{n+m}$$
$$\lambda_j\lambda_z \dots \lambda_w + \lambda_1\lambda_2 \dots \lambda_{(n+m)} \int_0^t \mathrm{d}\tau_{(m-1)}$$
$$\int_0^{\tau_{(m-1)}} \mathrm{d}\tau_{(m-2)} \dots \int_0^{\tau_1} e(\tau_0)\mathrm{d}\tau_0 = 0.$$

(62)

According to the previous method, we can obtain (62).

Additionally, when $\lambda_1 \gg 0$, $\lambda_2 \gg 0$, $\dots$, $\lambda_{m+n} \gg 0$ and $t \to \infty$, the derived equations are equivalent.

$$\begin{cases} \int_0^t \mathrm{d}\tau_{(m-1)} \int_0^{\tau_{(m-1)}} \mathrm{d}\tau_{(m-2)} \dots \int_0^{\tau_1} e(\tau_0)\mathrm{d}\tau_0 = 0, \\[4pt] \int_0^t \mathrm{d}\tau_{(m-2)} \int_0^{\tau_{(m-2)}} \mathrm{d}\tau_{(m-3)} \dots \int_0^{\tau_1} e(\tau_0)\mathrm{d}\tau_0 + \lambda_1 \\ \int_0^t \mathrm{d}\tau_{(m-1)} \int_0^{\tau_{(n-1)}} \mathrm{d}\tau_{(m-2)} \dots \int_0^{\tau_1} e(\tau_0)\mathrm{d}\tau_0 = 0, \\[4pt] \int_0^t \mathrm{d}\tau_{(m-3)} \int_0^{\tau_{(m-3)}} \mathrm{d}\tau_{(m-4)} \dots \int_0^{\tau_1} e(\tau_0)\mathrm{d}\tau_0 + \\ (\lambda_1 + \lambda_2) \int_0^t \mathrm{d}\tau_{(m-2)} \int_0^{\tau_{(m-2)}} \mathrm{d}\tau_{(m-3)} \dots \int_0^{\tau_1} \\ e(\tau_0)d(\tau_0) + \lambda_1\lambda_2 \int_0^t \mathrm{d}\tau_{(m-1)} \int_0^{\tau_{(m-1)}} \mathrm{d}\tau_{(m-2)} \\ \dots \int_0^{\tau_1} e(\tau_0)\mathrm{d}\tau_0 = 0, \\[4pt] \int_0^t \mathrm{d}\tau_{(m-4)} \int_0^{\tau_{(m-4)}} \mathrm{d}\tau_{(m-3)} \dots \int_0^{\tau_1} e(\tau_0)\mathrm{d}\tau_0 + \\ (\lambda_1 + \lambda_2 + \lambda_3) \int_0^t \mathrm{d}\tau_{(m-3)} \int_0^{\tau_{(m-3)}} \mathrm{d}\tau_{(m-4)} \\ \dots \int_0^{\tau_1} e(\tau_0)\mathrm{d}\tau_0 + (\lambda_1\lambda_2 + \lambda_1\lambda_3 + \lambda_2\lambda_3) \\ \int_0^t \mathrm{d}\tau_{(m-2)} \int_0^{\tau_{(m-2)}} \mathrm{d}\tau_{(m-1)} \dots \int_0^{\tau_1} e(\tau_0)d\tau_0 \\ +\lambda_1\lambda_2\lambda_3 \int_0^t \mathrm{d}\tau_{(m-1)} \int_0^{\tau_{(m-1)}} \mathrm{d}\tau_{(m-2)} \\ \dots \int_0^{\tau_1} e(\tau_0)\mathrm{d}\tau_0 = 0, \\[4pt] e(t) + \sum_{i=1}^{m-1}(\int_0^t \mathrm{d}\tau_{(m-1)} \int_0^{\tau_{(m-1)}} \mathrm{d}\tau_{(m-2)} \dots \\ \int_0^{\tau_1} e(\tau_0)d\tau_0)^{(i)} \sum_{j=1}^m \sum_{z=j+1}^m \dots \sum_{w=m-i}^m \\ \lambda_j\lambda_z \dots \lambda_w + \lambda_1\lambda_2 \dots \lambda_m \int_0^t \int_0^{\tau_{(m-1)}} \dots \\ \int_0^{\tau_1} e(\tau_0)\mathrm{d}\tau_0 = 0, \\[4pt] e(t)^{(n)} + \sum_{i=1}^{n+m-1}(\int_0^t \mathrm{d}\tau_{(m-1)} \int_0^{\tau_{(m-1)}} \mathrm{d}\tau_{(m-2)} \\ \dots \int_0^{\tau_1} e(\tau_0)d\tau_0)^{(i)}) \sum_{j=1}^{n+m} \sum_{z=j+1}^{n+m} \dots \\ \sum_{w=n+m-i}^{n+m} \lambda_j\lambda_z \dots \lambda_w + \lambda_1\lambda_2 \dots \lambda_{(n+m)} \\ \int_0^t \mathrm{d}\tau_{(m-1)} \int_0^{\tau_{(m-1)}} \mathrm{d}\tau_{(m-2)} \dots \int_0^{\tau_1} e(\tau_0)\mathrm{d}\tau_0 = 0. \end{cases}$$

## 11  $n$-Times-Integral ZE inequations of UPV situation

In this section, we discuss $n$-Times-integral ZE inequations of UPV. We also derive and analyze the formulas pertaining to $n$-Times-Integral of various orders, specifically $-n$, $-(n-1)$, $-(n-2)$, $-(n-3)$ and 0.

1) $(-n)$-order $n$-Integral General Formula:

$$\int_0^t \int_0^{\tau_{(n-1)}} \dots \int_0^{\tau_1} e(\tau_0)\mathrm{d}\tau_0 \le 0. \qquad (63)$$

In the context of $t \gg 0$, we define $e(t)$ in $e(t) \le 0$ as $\int_0^t \int_0^{\tau_{(n-1)}} \dots \int_0^{\tau_1} e(\tau_0)\mathrm{d}\tau_0$.

2) $(1-n)$-order $n$-Integral General Formula:
By defining $e(t)$ in $\dot{e}(t) + \lambda_1 e(t) \leq 0$ as $\int_0^t \int_0^{\tau(n-1)} \ldots \int_0^{\tau_1} e(\tau_0) d\tau_0$, we arrive at (63) when $\lambda_1 \gg 0$ and $t \to \infty$.

$$\int_0^t \int_0^{\tau(n-2)} \ldots \int_0^{\tau_1} e(\tau_0) d\tau_0 + \lambda_1$$
$$\int_0^t \int_0^{\tau(n-1)} \ldots \int_0^{\tau_1} e(\tau_0) d\tau_0 \leq 0. \tag{64}$$

According to ZE, we arrive at the (66) when $\lambda_1 \gg 0$ and $t \to \infty$.

3) $(2-n)$-order $n$-Times-Integral General Formula:
By defining $e(t)$ in $\dot{e}(t) + \lambda_2 e(t)$ as $\int_0^t \int_0^{\tau(n-2)} \ldots \int_0^{\tau_1} e(\tau_0) d\tau_0 + \lambda_1 \int_0^t \int_0^{\tau(n-1)} \ldots \int_0^{\tau_1} e(\tau_0) d\tau_0$, we arrive at the following formula when $\lambda_1 \gg 0$, $\lambda_2 \gg 0$ and $t \to \infty$:

$$\int_0^t \int_0^{\tau(n-3)} \ldots \int_0^{\tau_1} e(\tau_0) d\tau_0 + (\lambda_1 + \lambda_2) \int_0^t \int_0^{\tau(n-2)}$$
$$\ldots \int_0^{\tau_1} e(\tau_0) d(\tau_0) + \lambda_1 \lambda_2 \int_0^t \int_0^{\tau(n-1)} \ldots$$
$$\int_0^{\tau_1} e(\tau_0) d\tau_0 \leq 0. \tag{65}$$

4) $(3-n)$-order $n$-Times-Integral General Formula:
By defining $e(t)$ in $\dot{e}(t) + \lambda_3 e(t)$ as (64), we arrive at the following formula when $\lambda_1 \gg 0$, $\lambda_2 \gg 0$, $\lambda_3 \gg 0$ and $t \to \infty$:

$$\int_0^t \int_0^{\tau(n-4)} \ldots \int_0^{\tau_1} e(\tau_0) d\tau_0 + (\lambda_1 + \lambda_2 + \lambda_3)$$
$$\int_0^t \int_0^{\tau(n-3)} \ldots \int_0^{\tau_1} e(\tau_0) d\tau_0 + (\lambda_1 \lambda_2 + \lambda_1 \lambda_3 +$$
$$\lambda_2 \lambda_3) \int_0^t \int_0^{\tau(n-2)} \ldots \int_0^{\tau_1} e(\tau_0) d\tau_0 + \lambda_1 \lambda_2 \lambda_3$$
$$\int_0^t \int_0^{\tau(n-1)} \ldots \int_0^{\tau_1} e(\tau_0) d\tau_0 \leq 0. \tag{66}$$

5) 0-order $n$-Times-Integral General Formula:

$$e(t) + \sum_{i=1}^{n-1} \left( \int_0^t \int_0^{\tau(n-1)} \ldots \int_0^{\tau_1} e(\tau_0) d\tau_0 \right)^{(i)} \sum_{j=1}^n$$
$$\sum_{z=j+1}^n \ldots \sum_{w=n-i}^n \lambda_j \lambda_z \ldots \lambda_w + \lambda_1 \lambda_2 \ldots \lambda_n$$
$$\int_0^t \int_0^{\tau(n-1)} \ldots \int_0^{\tau_1} e(\tau_0) d\tau_0 \leq 0. \tag{67}$$

We suppose that the formula is valid for the $(-1)$ order. By incorporating this into formula $\dot{e}(t) + \lambda_n e(t) \leq 0$, we can obtain (66). Moreover, when $d(t) = 0$, the formula for the 0-order derivative is

given by the following formula.

$$a(t) + \sum_{i=1}^{n-1} \left( \int_0^t \int_0^{\tau(n-1)} \ldots \int_0^{\tau_1} a(\tau_0) d\tau_0 \right)^{(i)} \sum_{j=1}^n$$
$$\sum_{z=j+1}^n \ldots \sum_{w=n-i}^n \lambda_j \lambda_z \ldots \lambda_w + \lambda_1 \lambda_2 \ldots \lambda_n$$
$$\int_0^t \int_0^{\tau(n-1)} \ldots \int_0^{\tau_1} a(\tau_0) d\tau_0 \leq 0.$$

Additionally, when $\lambda_1 \gg 0$, $\lambda_2 \gg 0$, $\ldots$, $\lambda_n \gg 0$ and $t \to \infty$, the derived inequations are equivalent.

$$\begin{cases} \int_0^t \int_0^{\tau(n-1)} \ldots \int_0^{\tau_1} e(\tau_0) d\tau_0 \leq 0, \\ \int_0^t \int_0^{\tau(n-2)} \ldots \int_0^{\tau_1} e(\tau_0) d\tau_0 + \lambda_1 \int_0^t \int_0^{\tau(n-1)} \ldots \int_0^{\tau_1} \\ e(\tau_0) d\tau_0 \leq 0, \\ \int_0^t \int_0^{\tau(n-3)} \ldots \int_0^{\tau_1} e(\tau_0) d\tau_0 + (\lambda_1 + \lambda_2) \int_0^t \int_0^{\tau(n-2)} \\ \ldots \int_0^{\tau_1} e(\tau_0) d(\tau_0) + \lambda_1 \lambda_2 \int_0^t \int_0^{\tau(n-1)} \ldots \\ \int_0^{\tau_1} e(\tau_0) d\tau_0 \leq 0, \\ \int_0^t \int_0^{\tau(n-4)} \ldots \int_0^{\tau_1} e(\tau_0) d\tau_0 + (\lambda_1 + \lambda_2 + \lambda_3) \\ \int_0^t \int_0^{\tau(n-3)} \ldots \int_0^{\tau_1} e(\tau_0) d\tau_0 + (\lambda_1 \lambda_2 + \lambda_1 \lambda_3 + \\ \lambda_2 \lambda_3) \int_0^t \int_0^{\tau(n-2)} \ldots \int_0^{\tau_1} e(\tau_0) d\tau_0 + \lambda_1 \lambda_2 \lambda_3 \\ \int_0^t \int_0^{\tau(n-1)} \ldots \int_0^{\tau_1} e(\tau_0) d\tau_0 \leq 0, \\ e(t) + \sum_{i=1}^{n-1} (\int_0^t \int_0^{\tau(n-1)} \ldots \int_0^{\tau_1} e(\tau_0) d\tau_0)^{(i)} \sum_{j=1}^n \\ \sum_{z=j+1}^n \ldots \sum_{w=n-i}^n \lambda_j \lambda_z \ldots \lambda_w + \lambda_1 \\ \ldots \lambda_n \int_0^t \int_0^{\tau(n-1)} \ldots \int_0^{\tau_1} e(\tau_0) d\tau_0 \leq 0. \end{cases}$$

## 12  Conclusion

In this paper, we have discussed the Zhang equivalency of integral type, including equalities and inequalities with unequal parameter values. We have established the validity of these formulas through the use of mathematical induction.

Moreover, building on the established formulas, we have readily observed the following formula.

$$\epsilon_n(t) = e^{(n)}(t) + \sum_{i=1}^{n-1} e^{(n)}(t) \left( \sum_{j=1}^n \sum_{z=j+1}^n \ldots \sum_{w=n-i}^n \lambda_j \ldots \right.$$
$$\left. \lambda_w \right) + \lambda_1 \ldots \lambda_n e(t) \tag{68}$$

In the case of equalities, it is necessary only to let $\epsilon_n(t) = 0$. Furthermore, $e(t)$ can be any general function, and it can also correspond to a first derivative, a second derivative, or even derivatives of higher orders. In the case of inequalities, it is necessary only to let $\epsilon_n(t) \leq 0$. Furthermore, $e(t)$ can be any general function, and it can also correspond to a first derivative, a second derivative, or even derivatives of higher orders. The two formulas stated above are both valid $\lambda_1 \gg$, $\lambda_2 \gg 0$, $\ldots$, $\lambda_n \gg 0$ and $t \to \infty$.

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
