# OpenReview forum: "Integral-Related ZE (Zhang-Equivalency) Equations and Inequations of Unequal-Parameter-Value (UPV) Situation"
_IEEE.org/ICIST/2024/Conference — IEEE ICIST 2024 Conference Submission_

### Official Review · Reviewer_c9Jo · 2024-08-21
**Integral-Related ZE (Zhang-Equivalency) Equations and Inequations of Unequal-Parameter-Value (UPV) Situation**

**Rating:** 7
**Confidence:** 2

**Review:**

In this paper, the authors have discussed the Zhang equivalency of integral type, including equalities and inequalities with unequal parameter values. The authors have established the validity of these formulas through the use of mathematical induction. However, this paper seems to focus on theoretical analysis. It would be best if the authors could add some simulation results to validate the effectiveness of the method. In addition, the research motivation needs further clarification.

---

### Official Review · Reviewer_tGxj · 2024-08-23
**This paper can be accepted after minor modifications**

**Rating:** 7
**Confidence:** 4

**Review:**

This paper elaborates on the notion of Zhang equivalency (ZE) and explores the scenarios of equations and inequations within the context of the unequal-parameter-value (UPV) situation. Utilizing the framework of ZE, this paper conducts a thorough examination of the equations and inequations pertaining to once, twice, thrice, and n-times. Furthermore, this paper arrives at the general integral-related ZE formulas.
1). The main motivation of the proposed scheme should be described more clearly.
2). The hierarchy of the article is unclear, and the structure needs to be reconstructed.

---

### Official Review · Reviewer_Szyg · 2024-08-25
**In this paper, the authors conduct a thorough examination of equations and inequalities of the 1st, 2nd, 3rd, and n degrees. Further, the authors obtained the general ZE formula related to the integral. However。 1. What assumptions and lemma are used in theoretical derivation, and explain the reasonableness? 2. Emphasize when original work.**

**Rating:** 7
**Confidence:** 4

**Review:**

In this article, the authors provide a detailed theoretical analysis that elaborates on the concept of tension equivalence (ZE) and then discusses the case of inequalities in the case of equations and parameter values are not equal (UPV). Using ZE's framework, the authors thoroughly studied equations and inequalities for 1, 2, 3, and n degrees. The authors then get the general ZE formula related to the integral.
However. 1. Please describe the assumptions and lemma used in the theoretical derivation and explain the rationale.
2. Emphasize the original work of the paper.

---

### Decision · Program_Chairs · 2024-09-08

Accept (Oral)